# Transcatheter Aortic Valve Implantation in Patients with Previous Mitral Valve Surgery—Review

**DOI:** 10.3390/jcm14030735

**Published:** 2025-01-23

**Authors:** Anna Olasińska-Wiśniewska, Marcin Misterski, Marek Grygier, Janusz Konstanty-Kalandyk, Tomasz Urbanowicz, Maciej Lesiak, Marek Jemielity, Maciej Dąbrowski

**Affiliations:** 1Department of Cardiac Surgery and Transplantology, Poznan University of Medical Sciences, 61-848 Poznan, Poland; mister@poczta.onet.pl (M.M.); turbanowicz@ump.edu.pl (T.U.); mjemielity@poczta.onet.pl (M.J.); 2I Department of Cardiology, Poznan University of Medical Sciences, 61-848 Poznan, Poland; marek.grygier@skpp.edu.pl (M.G.); maciej.lesiak@skpp.edu.pl (M.L.); 3Clinical Department of Cardiac Surgery and Transplantation, St. John Paul II Hospital, 31-202 Kraków, Poland; jakonstanty@poczta.onet.pl; 4Department of Interventional Cardiology & Angiology, National Institute of Cardiology, 04-628 Warsaw, Poland

**Keywords:** transcatheter, aortic stenosis, mitral prosthesis

## Abstract

Transcatheter aortic valve implantation (TAVI) has become an optimal alternative in selected groups of patients and evolved from procedures in non-option patients to lower-risk-profile patients. One of its main indications is previous cardiac surgery, since redo-intervention is burdened with a higher risk of complications. However, TAVI after mitral valve surgery may raise concerns due to potential interference with the mitral prosthesis or ring during or after the procedure. The present paper reviews the current knowledge, including possible complications and procedural aspects.

## 1. Introduction

Calcific aortic stenosis (AS) is the most common degenerative valvular disease in high-income countries. Its prevalence is higher with age. Echocardiography represents the main diagnostic tool for aortic valve evaluation [1]. With the population aging, the percentage of elderly patients requiring therapy is significant, and the complexity of decision making increases. Risk stratification among patients with different stages of aortic stenosis advancement provides key information for the timing of interventions [2]. Stress echocardiography may be helpful in more challenging cases [3,4], such as severe septal hypertrophy, to guide decision making. While optimal medical therapy is implemented for coronary artery disease [5], currently, there is no established pharmacotherapy to halt or reverse the progression of aortic stenosis [6], despite numerous promising drugs [7,8]. Surgical aortic valve replacement (SAVR) is a recommended method of treatment in the majority of younger AS patients without substantial peri-operative risk, while elderly subjects and those burdened with severe co-morbidities or frailty are often deferred from surgery. Recently, transcatheter aortic valve implantation (TAVI) has become an optimal alternative in selected groups of patients, evolving from procedures dedicated to non-operative patients to those at a moderate or severe surgical risk and those over 75 years old at a low risk [9,10]. Since indications for TAVI have changed to the lower-risk and younger populations, the matter of prosthesis durability is noteworthy. Recently announced trials presented satisfactory TAVI results, comparable to SAVR in low-risk patients [11,12,13]. However, there are still several uncertainties, as TAVI patients are often burdened with co-morbidities such as diabetes, endocrine disorders, kidney disease, and hyperlipidemia, which may additionally influence valve durability and performance. Hemodynamic valve deterioration is associated with a 5-fold increased risk of repeat aortic valve intervention [14]. However, its incidence rate is unknown in patients with previous mitral surgery. Undoubtedly, the rapid advancement in prostheses and delivery system design has resulted in a reduced risk of peri-procedural complications. This includes lower rates of paravalvular leak, pacemaker implantation, bleeding, and vascular complications. Local compared to general anesthesia offers substantial advantages, such as a reduced intensive care unit stay and early mobilization [15,16]. Additionally, there has been a shortening of hospital stays and a shift toward more liberal post-procedural pharmacotherapy, with a focus on single antiplatelet use. Moreover, patients with more advanced co-morbidities and a poor general health status are less limited regarding qualification for the procedure.

## 2. Previous Cardiac Surgery

A considerable number of patients who require cardiac surgery for rheumatic heart disease present mild aortic valve disease at the time of mitral intervention [17]. However, most do not progress to severe disease in the long-term follow-up. Previous cardiac surgery is one of the reasons for the disqualification of elderly patients due to an increased perioperative risk of morbidity and mortality, as such patients are usually older and have more advanced co-morbidities compared to primary surgery ones. The prevalence of concomitant coronary artery disease is high and ranges between 52 and 68% [18,19,20]. A significant number of patients have previously undergone coronary artery bypass grafting (CABG) or other types of cardiac surgery. Redo surgery carries a considerably higher operative risk due to its procedural complexity, the potential for bypass injury when crossing the chest midline, and the presence of adhesions between cardiac structures and the chest wall [21]. The meta-analysis by Bajaj et al. [22] of 23 studies with 18,023 patients undergoing TAVI, including 4441 who had any previous cardiac surgery, showed good clinical outcomes in terms of morbidity and mortality, and complication rate results similar to those of patients without a history of surgical intervention. The 1-year mortality was 17.4% in patients who underwent surgery, while it was 18.7% in the non-cardiac surgery group. There were, however, no subgroups of patients who underwent only mitral surgery. Out of the 23 studies, 15 included previous CABG and 8 CABG and valve surgeries. Previous cardiac surgery requiring the opening of the pericardium has been included in the EuroScore II calculator as a risk factor for cardiac surgical mortality [23]. The main advantage of TAVI in this population is its minimally invasive character compared to SAVR, regardless of being obtained with ministernotomy or right anterior mini thoracotomy. Patients’ preferences concerning redo surgery must be considered.

The first large randomized clinical TAVI trials excluded patients with prosthetic valves in any position [24,25]. Though several concerns have been raised, this population of patients has garnered interest, as re-operation is burdened with significant perioperative risk. First, a 67-year-old TAVI patient after mitral valve replacement (MVR) and coronary bypass grafting revascularization was described by Rodes-Cabau et al. [26] by transapical access, with optimal procedural effects without complications. Therefore, subsequent operators performed the procedure and published their outcomes, providing the procedural “tips and tricks”. In two multicenter registries, the procedural success was high, ranging from 97.4 to 98.6%, with a device success of 72.2–86.3% and a mortality rate that was comparable to other TAVI subpopulations [27,28]. Several methods have been implemented to improve the outcomes of this specific type of procedure.

## 3. TAVI After Mitral Valve Surgery—Concerns and Perioperative Techniques

The primary avoidance of such a group of patients resulted from several concerns about rigid mechanical prostheses. The main worry was the suspicion of possible mitral prosthesis dysfunction or aortic bioprosthesis embolization during valve positioning and deployment. This complication occurred in 6.7% of patients included in the registry by Amat-Santos et al. [27]. The presence of a previously implanted mitral prosthesis may interfere with the aorto-mitral space, causing its reduction or absence. The prosthesis deployment may be limited, leading to inadequate prosthesis expansion or embolization [29] in the mechanism of slipping, called a watermelon seed effect. Among the predisposing risk factors, a large native aortic annulus with a low calcific burden, predominant aortic insufficiency, and a short aorto-mitral diameter have been mentioned [30]. Though a strong correlation between prosthesis embolization and an aorto-mitral distance of less than 7 mm has been postulated [27], several consecutive papers have reported safe procedures without embolization, even with a much shorter space (see Table 1). Importantly, prosthesis dislocation may not necessarily occur during the procedure. Episodes of late embolization have been described, raising operators’ awareness about long-term observation and prompt diagnostics implementation in cases of sudden deterioration. Baumbach et al. [31] presented a prosthesis dislocated into the left ventricle on the sixth post-procedural day. The stiff annulus of the mitral valve prosthesis caused a non-circular aortic annulus in the region of noncoronary sinus and left-to-noncoronary commissure. The patient underwent redo surgery, complicated with the need for ascending aorta replacement and recurrent bleeding. Maroto et al. [32] described aortic bioprosthesis dislocation revealed three weeks after the procedure.

Progress in pre-procedural imaging is crucial in optimizing procedural planning and reducing the risk of complications, such as prosthesis embolization or left ventricular outflow tract obstruction. The use of pre-procedural computed tomography (CT) imaging will help operators to optimize the procedure by maximizing prosthesis hemodynamic function and durability [33]. Chao et al. [29] underlined gated cardiac computed tomography (CT) angiography’s beneficial role in assessing the aorto-mitral distance available for prosthesis expansion before the procedure. The authors also evaluated post-operative results with CT and revealed the compression of the fully expanded prosthesis stent on the surrounding tissue. The newest reports have shown artificial intelligence (AI) implementation in pre-procedural planning [30]. AI simulation models using patient-specific computer tomography reconstruction provide a tailored computer simulation of the TAVI procedure, enabling the prediction of device–anatomy interaction and the risk of complications such as paravalvular leak or coronary obstruction.

Intra-procedural multidisciplinary guidance with echocardiography and fluoroscopy is crucial in the assessment of interference between aortic bioprosthesis and mitral prosthesis and for adjusting the extent of stent protrusion into the left ventricular outflow tract (LVOT) [34]. Asil et al. [35] proposed transesophageal echocardiography. However, the majority of current procedures are performed without this type of imaging, relying on transthoracic echocardiography and fluoroscopy to avoid the need for general anesthesia [36]. Based on a CT analysis, a suitable projection for fluoroscopy should be chosen to expose the aorto-mitral distance to facilitate the correct positioning of the prosthesis.

The chronic anticoagulation requirement for mechanical prostheses may increase the risk of bleeding complications. Published reports have described antithrombotic regimens, including the withdrawal of oral anticoagulation at least 48 h before the procedure, the introduction of low-molecular-weight heparin before the procedure, and restarting oral anticoagulation within the first 24 h, which enabled avoiding the risk of prosthesis thrombosis or major bleeding [37].

The type of mitral prosthesis should raise technical pre-procedural caution. Mechanical prostheses have a rigid cage with or without protruding pivot guards, while biological ones have more prominent commissural struts. Mitral bioprosthetic valves are associated with a higher risk of interference from the frame of the aortic prosthesis with the mitral valve struts, which may protrude more into the left ventricular outflow tract than low-profile mechanical prostheses [38,39,40,41]. Soon et al. [42] described cases of balloon shifts during valvuloplasty. During balloon postdilatation, attention should also be given to the potential risk of balloon interference with the mitral prosthesis. In the case of balloon postdilatation, asymmetrical balloon positioning and adjustment towards the aorta may be considered.

There are opposing opinions regarding the superiority of ballon-expandable or self-expanding bioprostheses. During the implantation of balloon-expandable TAVI prostheses, attention should also be given to the potential risk of balloon interference with the mitral prosthesis. Bagur et al. [43] pointed out that the instability of balloon expansion may increase the risk of malposition and embolization. Chao et al. [29] suggested, with caution, using a balloon-expandable prosthesis in patients with small aortic roots, resulting in significant prosthesis oversizing. Importantly, first-generation self-expanding bioprostheses were longer and could protrude into the LVOT, thus interfering with the stiff mitral cage [43]. An incomplete inflow expansion or difficulties with mitral prosthesis opening could be enhanced. The behavior of the balloon should be monitored during aortic valvuloplasty, with caution as to how it inflates, its potential displacement, and any residual waist [35]. Aggressive valve oversizing should be avoided to diminish the risk of prosthesis displacement [35,40]. Asil et al. [35] summarized the prosthesis options depending on the aorto-mitral space. They proposed that a distance of 4 mm should be required to permit the secure deployment of a self-expanding frame without interfering with the mitral prosthesis leaflets. If the aorto-mitral distance is less than 4 mm, balloon-expandable prostheses should be opted for, as these types have to align with the aortic valve annulus, limiting the chance of interference with the mitral prosthesis. The advantage of self-expanding prostheses results from a gradual deployment process, which permits the adequate adjustment of implantation depth. In cases of aortic bioprostheses extending too deeply into the LVOT, maneuvers of valve withdrawal into the ascending aorta before total release or bail-out repositioning valves after release have been proposed and described [35]. Short-frame TAVI prostheses have been postulated by other authors [44,45,46] due to a lower valve length and a margin that does not reach more than 2 mm into the LVOT below the aortic annulus, offering a reasonably safe distance between the two prostheses. A shorter valve length may prevent asymmetrical deployment [46]. Technical aspects include the active clip fixation of the native valve, thereby reducing radial forces on the nearby tissues [44].

An important advantage of the currently used prostheses is the possibility of recapturing and repositioning the device during deployment. Undoubtedly, the operators’ experience with a particular prosthesis type is of the utmost importance.

Femoral access is recommended according to current guidelines [9], and was predominantly used in the presented cases. However, some authors have underlined the benefits of other approaches. Beller et al. [38] argued that the mechanism for positioning with transapical access might be better controlled compared with femoral access, considering the short aorto-mitral distance. Drews et al. [47] postulated the transapical approach as a safe one, while transcatheter wires should be introduced carefully to avoid touching the mitral prosthesis. Additionally, they recommended that prosthesis positioning and liberation should be performed under simultaneous angiography with contrast media to find the optimal position and reduce the risk of paravalvular leaks and contact with the mitral prosthesis. However, currently, transapical access is generally avoided. Bruschi et al. recommended a distal axillary approach, such as subclavian, as it may provide a closer TAVI prosthesis placement and high deployment control [48]. Moreover, numerous operators have underlined the significance of the possibility of resheathing and redeploying the aortic prosthesis [34,48]. Therefore, in the case of intra-procedural signs of acute mitral prosthesis dysfunction during transcatheter prosthesis deployment, the latter may be repositioned [49] and gradually released with less impact on the aortic root [30].

Avoiding cardiopulmonary bypass (CPB) is one of the main advantages of TAVI. However, TAVI in patients with a mitral valve prosthesis may be related to an increased risk of hemodynamic collapse. In severely depressed left ventricular ejection fraction patients, especially those in cardiogenic shock, cardiopulmonary bypass (CPB) [39,50] support without cardiac arrest and extracorporeal membrane oxygenation (ECMO) [51] have been proposed during the TAVI procedure to optimize its safety. It may be beneficial in patients with severe left and right ventricular dysfunction with pulmonary hypertension who may not tolerate rapid pacing during the valvuloplasty procedure or prosthesis implantation, resulting in hemodynamic collapse [39].

While with time and operators’ experience, TAVI has gained approval in patients with previous MVR, even more challenging high-risk surgical circumstances may occur, such as the need for intervention for severe multiple valvular heart disease. Lima et al. [40] described successive procedures of concomitant TAVI and tricuspid valve-in-valve in an elderly woman with prior mitral mechanical prosthesis and tricuspid bioprosthesis surgical implantation. The authors concluded that special attention should be paid to appropriate pre-procedural planning, as well as post-procedural antithrombotic therapy, as the patient developed tricuspid bioprosthesis thrombosis due to a subtherapeutic international ratio (INR) in the four-month follow-up.

**Table 1 jcm-14-00735-t001:** Literature review.

Author, Data	Patients Age, Sex	Surgical Risk	Type of Mitral Prosthesis	TAVI Prosthesis	Access	Outcome	Peri-Procedural Complications	Aorto-Mitral Diameter
Rodes-Cabau et al., 2008 [26]	67, M	STS 7.5%	St Jude Medical	Edwards Sapien 26 mm	Transapical	MeanGrad 12 mmHg, no PVL	No	NA
Maroto et al., 2009 [32]	75, F	LogisticES 29.5%	Bileaflet mechanical MVP	Edwards Sapien 23 mm	Transapical	Mild PVL, in 3-week FU—dyspnea due to bioprosthesis displacement treated with SAVR complicated by a hemispheric cerebrovascular accident	No	NA
Scherner et al., 2009 [52]	84, F	ES 35%, STS 24%	Bileaflet 29 mm	Edwards Sapien 26 mm	Transapical	MeanGrad 11 mmHG, optimal status at 2-month FU	No	NA
Bruschi et al., 2009 [53]	72, F	LogisticES 23% (range 23–44%), STS > 33%	Sorin allcarbon monodisc 31 mm	CoreValve 26 mm	Femoral	MeanGrad 9 mmHg, FU 4–12 months—MeanGrad 10 mmHg, asymptomatic, pts no 3- HTX	No	NA
	77, F		Sorin allcarbon monodisc 29 mm	CoreValve. 26 mm	Femoral		No	NA
	60, F		Sorin allcarbon monodisc 25 mm	CoreValve 26 mm	Femoral		No	NA
	77, F		Sorin bicarbon monodisc 29 mm	CoreValve 26 mm	Femoral		No	NA
Dumonteil et al., 2009 [54]	82, F	STS 26.5%	Lillehei-Kaster	Edwards Sapien 23 mm	Femoral	Mild PVL, 1 month FU NYHA II	No	9.7 mm
Chao et al., 2010 [29]	72, F	Logistic ES 15%, STS 5.3%	St. Jude Medical	ES 23 mm	Transapical	MeanGrad 11 mmHg, mild PVL On CT, fully expanded TAVI prosthesis with obliteration of the aorto-mitral space, NYHA II in 3 mFU	No	3.7 mm
Baumbach et al., 2011 [31]	82, F	ES 37%, STS 5.8%	Carpentier-Edwards 29 mm	Edwards Sapien 23 mm	Transapical	MeanGrad 8 mmHg	2 weeks after the procedure—aortic prosthesis dislocation, AVR via standard sternotomy complicated with ascending aorta replacement, and recurrent bleeding	NA
Beller et al., 2011 [38]	5 females Mean (SD) age of 80 ± 5.1	Mean LogisticES 39.3 ± 20.5%	-	ES 26 mm and 23 mm	4× apical 1× femoral	3 mild PVL	No access site complications, 2 respiratory failures, 1 AKI with hemofiltration, 2 deaths due to fulminant pneumonia	9–11 mm
García et al., 2011 [55]	71, M	ES 19.7%	ATS 29 mm	Edwards Sapien 26 mm	Femoral	30-day FU NYHA I	No	7.3 mm
	83, M	ES 38%	St Jude	Edwards Sapien 23 mm	Femoral	3-month FU NYHA I	No	7.3 mm
	74, M	ES 25%	St Jude	Edwards Sapien XT 26 mm	Femoral	Optimal status at 1 month FU NYHA II	Vascular complication with need for stent implantation, permanent pacemaker implantation	7 mm
Soon et al., 2011 [42]	86, F	LogisticES 70.43%, STS 18.7%	Bjork–Shiley (27 mm)	Edwards Sapien 23 mm	Femoral or transapical	Mild PVL	All prostheses successfully implanted	NA
	82, F	LogisticES 30.53%, STS 13.8%	St. Jude (25 mm)	Edwards Sapien 26 mm		-	-	
	78, M	LogisticES 32.62%, STS 5%	St. Jude (27 mm)	Edwards Sapien 26 mm		-	-	
	67, M	LogisticES 13.27%, STS 4.6%	St. Jude (25 mm)	Edwards Sapien 26 mm		Trivial PVL	-	
	77, F	LogisticES 16.13%, STS 8.2%	St. Jude (27 mm)	Edwards Sapien 26 mm		Mild PVL	Ventricular balloon shift by 2 mm, aortic prosthesis shift by 3 mm	
	71, F	LogisticES 11.42%, STS 4.6%	CARBOMEDIC 25 MM	Edwards Sapien 23 mm		Died by 144-day FU	Slight balloon displacement during valvuloplasty	
	82 F	LogisticES 13.03%, STS 8.9%	ST JUDE	Edwards Sapien 23 mm		Mild PVL		
	69, F	LogisticES 31.91%, STS 10.3%	PERIMOUNT 27 MM	Edwards Sapien 23 mm		Moderate PVL	Significant balloon displacement towards the aorta during inflation	
	76, M	LogisticES 38.75%, STS 15.5%	Mosaic	Edwards Sapien 26 mm		Trivial PVL	Significant balloon displacement towards the aorta during inflation, 1 failed due to gross balloon shift, valve embolization, she returned for TAVI 4 Y later	
	88, F	LogisticES 28.59%, STS 9.8%	Mosaic	Edwards Sapien		Trivial PVL		
Drews et al., 2011 [47]	82, F	ES 45%, STS 23%	Carpientier-Edwards Physio ring	Edwards Sapien 23 mm	Transapical	Proper function on ECHO, at 8-month FU IE and death	No	NA
	37, F	ES 85%, STS 75%	St Jude Medical 29 mm	Edwards Sapien 23 mm	Transapical	Mild PVL	Severe left HF, ECMo, death	NA
	75, M	ES 89%, STS 42%	Aortic homograft and Biological Hancock 33 mm	Edwards Sapien 26 mm	Transapical	No PVL, 1 Y FU well-functioning	No	NA
	80, F	ES 41%, STS 36%	Hancock 31 mm	Edwards Sapien 26 mm	Transapical	Proper aortic bioprosthesis function, no PVL 14-month FU well-functioning	No	NA
	82, F	ES 65%, STS 50%	Hancock 31 mm	Edwards Sapien 26 mm	Transapical	Trivial PVL	No	NA
	85, F	ES 45%, STS 32%	Bjork–Shiley	Edwards Sapien 23 mm	Transapical	Proper aortic bioprosthesis with minimal central leak, 2-month FU well-functioning	Transient dysfunction of the mitral prosthesis leaflet during introduction of the delivery system into the outflow tract of the left ventricle rescued by immediate temporary retraction of the delivery system	NA
Salinas et al., 2012 [56]	86, M	ES 43%	St Jude 27 mm	Edwards Sapien XT 23 mm	Femoral	Optimal status at 5 months FU, mean grad 10 mmhG	Balloon displacement and implantation with minor displacement	NA
Bruschi et al., 2013 [41]	4 pts described in 2009							
	74, F	STS 7.9%	Sorin allcarbon monodisc 27 mm	CoreValve 26 mm	Femoral	MeanGrad 8 mmHG, mild PVL, alive in 24 FU	No	NA
	31, M	STS 36.5% procedure in cardiogenic shock	Carpentier-Edwards 23 mm	CoreValve 26 mm	Femoral	MeanGrad 19 mmHG, mild PVL, after 60 days successful bridge to AVR + MVR	No	NA
	76, F	STS 6.1%	Edwards Physio ring 26 mm	CoreValve 29 mm	Femoral	MeanGrad 12 mmHG, mild PVL, alive in 17.9 months FU	No	NA
	72, M	STS 17.1%	ON-X bileaflet 25 mm	CoreValve 26 mm	Direct aorta	Mean Grad 9 mmHg, mild PVL, alive in 12 mFU	Pacemaker implantation due to AV block	NA
	83, F	STS 17.5%	Sorin bicarbon bileaflet 25 mm	CoreValve 26 mm	Direct aorta	MeanGrad 9 mmHG, no PVL, alive in 4 FU	No	NA
Vavuranakis et al., 2014 [57]	66, F	LogES 13.1	Omniscience	CoreValve 26 mm	Femoral	1-month FU meanGrad 5 mmHg, NYHA I	No	5.8 mm
	85, F	LogES 51.8%	St Jude	CoreValve 29 mm	Femoral	1 Y FU mean grad 4 mmHg, NYHA II	No	9.3 mm
O’Sullivan et al., 2015 [44]	60, F	ES II 6.48% STS 6.55%	Medtronic Hall disc valve	JenaValve 25 mm	Transapical	Grad 12 mmHg, no PVL	Left pleural effusion	4.8 mm
Mieres et al., 2015 [45]	83, F	ES II 23.1%, STS 50.2%	Mechanical MVP	JenaValve 23 mm	Transapical	Vmax 2.66 m/s, no PVL, 1 Y FU NYHA i-II	Low cardiac output and oliguria	NA
Bruschi et al., 2016 [48]	83, M	ESII 16.2%, STS 8%	Sorin bicarbon 27 mm	Portico 29 mm	Axillary	MeanGrad 8 mmHg, trivial PVL, on CT fully expanded TAVI prosthesis, normal opening of mitral prosthesis	No	
Asil et al., 2016 [35]	82, F	LogisticES 28%, STS 11%	Bioprosthesis	CoreValve 23 mm	Femoral surgical cut-down	MeanGrad 5 mmHg, no PVL, 11-month FU NYHA II	Vascular complication—femoral AV fistula, complete AV block—PPM	8 mm
	53, M	LogisticES 24%, STS 13%	MVR + CABG	CoreValve 29 mm	Femoral surgical cut-down	MeanGrad 12 mmHg, mild PVL, 19-month FU NYHA I	Vascular complication—femoral hematoma	6 mm
	72, M	LogisticES 47%, STS 8%	MVR	CoreValve 23 mm	Femoral Surgical cut-down	MeanGrad 3 mmHg, no PVL, 19-month FU NYHA II	No	5 mm
	76, F	LogisticES 43%, STS 11%	MVR	CoreValve Evolut R 29 mm	Femoral	MeanGrad 5 mmHg, mild PVL, 12-month FU NYHA I	No	5 mm
	68, F	LogisticES 24%, STS 12%	MVR	CoreValve 29 mm	Femoral Surgical cut-down	MeanGrad 10 mmHg, mild PVL, 12-month FU NYHA I	No	9 mm
	75, F	LogisticES 48%, STS 4%	MVR	CoreValve Evolut R 29 mm	Femoral	MeanGrad 6 mmHg, moderate PVL, 13-month FU NYHA II	Femoral pseudoaneurysm extravasation	6 mm
Wachter et al., 2016 [46]	76, M	ES II 11.61%, STS 5.52%	Carbomedics 27 mm	JenaValve 27 mm	Transapical	MeanGrad 10 mmHg, no PVL, FU 2.8 y	No	NA
	74, M	ES II 11.08%, STS 6.72%	Perimount Plus 27 mm	JenaValve 27 mm	Transapical	MeanGrad 14 mmHg, no PVL, FU 1.3 y	No	NA
Bagur et al., 2017 [43]	72,F	STS 6.1%	Bi-leaflet 31 mm	Acurate neo 25 mm	Femoral	MeanGrad 5 mmHg, no PVL, at 1 Y FU NYHA I-II	No	2.4 mm
Amat-Santos et al., 2017 [27]	Registry, 91 patients, mean 74.8 y, 71.4% F	Mean logisticES 27.43%, STS 8.88%	24 (26.4%) biological prostheses, 67 (73.6%) mechanical prostheses (19.4% monodisc, 80.6% bidisc)	51 ballon-expandable prosthesis (56%)	Femoral 79.1%	Device success, 72.2%, procedural success, 98.6%	TAVI device embolization in 6 (6.7%), need for second prosthesis in 5 (5.6%), permanent pacemaker in 12 (14.8%), stroke in 2 (2.5%), bleeding complications in 22 (24.2%)	
Korkmaz et al., 2018 [34]	53, M	LogES 12.8% STS 2.6%	Bileaflet mechanical	Portico 27 mm	Femoral	MeanGrad 8 mmHg, mild PVL, NYHA I	No	4.5–5 mm
Baldetti et al., 2019 [28]	OPTIMAL study, 154 patients, mean age 77.2 y, 79.9% F	Mean logistic ES 26.4%, mean STS 26.4%	Biological prosthesis in 47 (30.5%) and mechanical in 107 (69.5%)	Ballon expandable in 47.7%, self-expanding in 49%, lotus in 2.6%, and other design in 0.7%	Femoral, 77.9%, transapical, 15.7%, trans-subclavian, 2%, direct aorta, 3.9%, transcaval, 0.7%	Procedural success of 97.4%, device success of 86.3%, in follow-up, 2 late fatal mitral prosthesis thromboses and 1 fatal hemorrhagic stroke	Prostheses interference in 2 patients, with 1 complicated with TAVI prosthesis embolization, 4 (2.6%) with cerebrovascular accidents, 6.6% with major vascular and 14.4% with major bleeding complications, 5 in-hospital deaths	9.7 ± 4.8 mm
Li et al., 2019 [58]	67, F	ES 23.45% STS 8.073%	ON-X 25 mm	VenusA-Valve 23 mm	Femoral	MeanGRad 16.9 mmHG, no PVL, Asymptomatic at 6 month FU	No	7 mm
Chmielak et al., 2020 [36]	17 pts mean age 75 years	Mean ES II 8.7	St Jude Medical n = 9 pts, Medtronic Hall n = 3 pts, Sorin Bicarbon n = 1 pt, Carbomedics n = 1 pt, nonspecified n = 1 pt	CoreValve EVOLUT R (N = 7) CoreValve (n = 4) Sapien XT (n = 3) Accurate (n = 1)	Femoral, 1 subclavian	MeanGrad 38.3 mmHG, in FU—1 IE, 1—stenocardia with PCI chimney stenting of LM, 2 deaths (bleeding, sepsis)	1 cardiac tamponade treated with pericardiocentesis, 1 prosthesis implanted above the coronary ostia	NA
Guleria et al., 2022 [49]	61, M	ES 11%	ATS 29	CV Evolut R 34 mm	Femoral	MeanGrad 4 mmHg, no PVL	No	
Tébar Márquez et al., 2022 [37]	79, F	STS 2.87%	Sorin bicarbon 25 mm	Allegra 23, 27 or 31 mm	Femoral	MeanGrad 7 mmHg, no PVL ≥ 2	No	2 mm
	77, F	STS 4.68%	Sorin bicarbon 27 mm		Femoral	MeanGrad 4 mHg, no PVL ≥ 2	No	3.9 mm
	70, F	STS 3.2%	Medtronic Open Pivot 27 mm		Femoral	MeanGrad 6 mmHg, no PVL ≥ 2	No	5 mm
	68, F	STS 2.47%	Edwards MIRA 25 mm		Femoral	MeanGrad 7.5 mmHg, no PVL ≥ 2	No	5.5 mm
	78, F	STS 2.76%	Sorin bicarbon 25 mm		Femoral	MeanGrad 6 mmHg, no PVL ≥ 2	No	5 mm
	75, F	STS 2.9%	Medtronic Open Pivot 27 mm		Femoral	MeanGrad 10.5 mmHg, no PVL ≥ 2	No	3 mm
	85, M	STS 2.1%	Sorin bicarbon 27 mm		Femoral	MeanGrad 16 mmHg, no PVL ≥ 2	AV block, PPM	4 mm
	80, F	STS 4.51%	Sorin bicarbon 27 mm		Femoral	MeanGrad 8 mmHg, no PVL ≥ 2	AV block, PPM	2 mm
Endo et al., 2023 [39]	90, F	ESII 7.37% STS 15.8%	Carpentier–Edwards Perimount	Edwards Sapien 3 23 mm with CPB support	Femoral	MeanGrad 15 mmHG, np PVL, NYHA I	No	5.4 mm
Maiani et al., 2024 [30]	83, F	NA	Carbomedics 29 mm	CoreValve Evolut PRO Plus 26 mm	Femoral	Optimal result on CT	No	4.7 mm
	78, F	NA	SJM 29 mm	CoreValve Evolut PRO Plus 23 mm	Femoral	Optimal result on angiogram, no PVL	No	3.5 mm
	79, F	NA	Carbomedics 25 mm	CoreValve Evolut PRO plus 26 mm	Femoral	Optimal result on CT, no PVL	No	4.7 mm
	85, F	NA	Carbomedics 27 mm	CoreValve Evolut PRO Plus 26 mm	Femoral	Optimal result on CT, no PVL	No	7.5 mm
	80, M	NA	Carbomedics 29 mm	CoreValve Evolut PRO Plus 29 mm	Femoral	Optimal result on CT, no PVL	No	5.7 mm
	73, F	NA	Carbomedics 29 mm	CoreValve Evolut PRO Plus 29 mm	Femoral	Optimal result, no PVL	No	5 mm
Lima et al., 2024 [40]	83, F	ES II 14.2%	Monoleaflet Bjork–Shiley and tricuspid annuloplasty, followed by tricuspid Carpentier-Edwards 29 mm implantation	Navitor FlexNav 25 mm and Edwards Sapien 3 Ultra 26 mm in tricuspid bioprosthesis	Femoral	NYHA II, 4-month FU—symptomatic TVIV thrombosis treated with unfractionated heparin	No immediate periprocedural	2.8 mm

AV—atrioventricular, CT—computed tomography, ES—EuroScore, FU—follow-up, MVR—mitral valve replacement, NA—not available, NYHA—New York Heart Association, PPM—permanent pacemaker, STS—Society of Thoracic Surgeons, TAVI—transcatheter aortic valve implantation.

## 4. Limitation

Though we reviewed the majority of the literature concerning the issue of TAVI in patients after mitral surgery, we could have missed some of the published reports. However, we aimed to present the most important pre-procedural and peri-procedural concerns and management options. The lack of long-term data indicates the importance of registries and randomized controlled trials focusing on patients with prior mitral valve surgery undergoing TAVI.

## 5. Conclusions

Despite initial concerns, TAVI has occurred as a safe and effective method of treatment in patients with severe AS with previously implanted mitral prostheses and may be performed without an outstanding risk of complications. Proper pre-procedural planning with computed tomography, the chosen access type, and a precise evaluation of patient history, including the type of mitral prosthesis, pharmacotherapy, and cautious choice of aortic bioprosthesis, enable the optimization of outcomes.

The main weakness of such a conclusion is the limited data published, based mainly on case reports or case series, without long-term observation, which may lead to underestimating the procedural risks and follow-up effects. Nation-wide registries with long-term evaluations are necessary.

## Data Availability

Not applicable.

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
