# Peer review of "Transcatheter Aortic Valve Implantation in Patients with Previous Mitral Valve Surgery—Review"

_jcm, 2025, doi:10.3390/jcm14030735_

Round 1

Reviewer 1 Report

Comments and Suggestions for Authors

The manuscript provides a comprehensive review of TAVI in patients with prior mitral valve surgery. It effectively highlights procedural techniques, risks, and outcomes. The following points should be resolved:

While advancements in TAVI are outlined, the lack of pharmacological or non-invasive options to halt or reverse the progression of aortic stenosis provides a compelling argument for TAVI's evolution as a primary intervention. Add a sentence discussing how, unlike other cardiovascular conditions, aortic stenosis lacks disease-modifying medical treatments, thereby necessitating surgical or transcatheter solutions. I would reference this recent study 10.3390/jcm13216411.

Expand on the advantages and disadvantages of self-expanding vs. balloon-expandable prostheses in patients with prior mitral valve surgery. For example, clarify when a short-frame prosthesis is most appropriate.

Discuss the evolving role of pre-procedural imaging (e.g., gated cardiac CT and AI simulation models). Highlight how these techniques optimize procedural planning and reduce complications such as prosthesis embolization or LVOT obstruction.

Since TAVI indications are expanding to lower-risk and younger populations, discuss how durability concerns may differ in these groups, particularly those with prior mitral valve surgery.

While the paper acknowledges the lack of long-term data, it could further stress the importance of registries and randomized controlled trials focusing on patients with prior mitral valve surgery undergoing TAVI. Emphasize this gap as a critical area for future exploration.

Author Response

The manuscript provides a comprehensive review of TAVI in patients with prior mitral valve surgery. It effectively highlights procedural techniques, risks, and outcomes.

Dear Reviewer, thank you for your valuable comments. We corrected the manuscript according to your suggestions.

The following points should be resolved:

While advancements in TAVI are outlined, the lack of pharmacological or non-invasive options to halt or reverse the progression of aortic stenosis provides a compelling argument for TAVI's evolution as a primary intervention. Add a sentence discussing how, unlike other cardiovascular conditions, aortic stenosis lacks disease-modifying medical treatments, thereby necessitating surgical or transcatheter solutions. I would reference this recent study 10.3390/jcm13216411.

Dear Reviewer, thank you for the suggestion, we corrected the manuscript accordingly.

Expand on the advantages and disadvantages of self-expanding vs. balloon-expandable prostheses in patients with prior mitral valve surgery. For example, clarify when a short-frame prosthesis is most appropriate.

Dear Reviewer, thank you for the suggestion, we corrected the manuscript accordingly.

Discuss the evolving role of pre-procedural imaging (e.g., gated cardiac CT and AI simulation models). Highlight how these techniques optimize procedural planning and reduce complications such as prosthesis embolization or LVOT obstruction.

Dear Reviewer, thank you for the suggestion, we corrected the manuscript accordingly.

Since TAVI indications are expanding to lower-risk and younger populations, discuss how durability concerns may differ in these groups, particularly those with prior mitral valve surgery.

Dear Reviewer, thank you for the suggestion, we corrected the manuscript accordingly.

While the paper acknowledges the lack of long-term data, it could further stress the importance of registries and randomized controlled trials focusing on patients with prior mitral valve surgery undergoing TAVI. Emphasize this gap as a critical area for future exploration.

Dear Reviewer, thank you for the suggestion, we corrected the manuscript accordingly.

Reviewer 2 Report

Comments and Suggestions for Authors

In this interesting review, the authors analyzed all literature data concerning TAVI treatment in patients who previously underwent mitral valve surgery.

The main determinant for the TAVI success is the aorto-mitral distance available for prosthesis expansion, without interfering with the mitral prosthesis leaflets.

The authors described the potential complications associated with TAVI procedure, particularly the risk of balloon interference with the mitral prosthesis. 

Among the pre-procedural examinations, the authors mentioned the CT angiography, that is useful for assessing the aorto-mitral distance available for prosthesis expansion.

Moreover, the authors discussed the importance of intra-procedural multidisciplinary guidance with echocardiography and fluoroscopy in the assessment of interference between aortic bioprosthesis and mitral prothesis.

The manuscript is weel written and each section clearly presented.

I congratulate the authors for their work.

I have only one suggestion for the authors.

At the end of the Discussion section, the authors could also discuss the diagnostic role of exercise stress echocardiography (ESE) in patients affected by aortic stenosis (AS). Indeed, literature data indicate that ESE may provide incremental prognostic information in AS patients (PMID: 39704221 and PMID: 26223986). Moreover, ESE may detect or unmask an intraventricular or LVOT pressure gradient, especially in patients with severe septal hypertrophy (PMID: 39307332). The intraventricular or LVOT pressure gradient may suggest a narrow aorto-mitral distance.

The authors could mention and discuss the suggested references and suggest the readers potential indications to ESE in AS patients.

Is there a role for ESE in AS patients candidates to TAVI? What do the authors recommend about the use of ESE in these patients?

Author Response

Dear Reviewer, thank you for your valuable comments. We corrected the manuscript according to your suggestions.

[...]I have only one suggestion for the authors.

At the end of the Discussion section, the authors could also discuss the diagnostic role of exercise stress echocardiography (ESE) in patients affected by aortic stenosis (AS). Indeed, literature data indicate that ESE may provide incremental prognostic information in AS patients (PMID: 39704221 and PMID: 26223986). Moreover, ESE may detect or unmask an intraventricular or LVOT pressure gradient, especially in patients with severe septal hypertrophy (PMID: 39307332). The intraventricular or LVOT pressure gradient may suggest a narrow aorto-mitral distance.

The authors could mention and discuss the suggested references and suggest the readers potential indications to ESE in AS patients.

Is there a role for ESE in AS patients candidates to TAVI? What do the authors recommend about the use of ESE in these patients?

Dear Reviewer, thank you for the suggestion, we corrected the manuscript accordingly. ESE may be particularly important in challenging scenarios. We added this mode of diagnostic tool in the introduction section.